# Analysis of Maternity Rights Perception: Impact of Maternal Care in Diverse Socio-Health Contexts

**DOI:** 10.3390/ejihpe15020010

**Published:** 2025-01-23

**Authors:** Claudia Susana Silva-Fernández, Paul Anthony Camacho, María de la Calle, Silvia M. Arribas, Eva Garrosa, David Ramiro-Cortijo

**Affiliations:** 1Department of Biological & Health Psychology, Faculty of Psychology, Universidad Autónoma de Madrid, 28049 Madrid, Spain; 2Centro de Investigaciones, Fundación Oftalmológica de Santander, Bucaramanga 680003, Colombia; 3Obstetric and Gynecology Service, Hospital Universitario La Paz, Universidad Autónoma de Madrid, 28049 Madrid, Spain; 4Department of Physiology, Faculty of Medicine, Universidad Autónoma de Madrid, 28029 Madrid, Spain; 5Instituto Universitario de Estudios de la Mujer (IUEM), Universidad Autónoma de Madrid, 28049 Madrid, Spain; 6Grupo de Investigación en Alimentación, Estrés Oxidativo y Salud Cardiovascular (FOSCH), Instituto de Investigación Sanitaria, Hospital Universitario La Paz (IdiPAZ), 28046 Madrid, Spain

**Keywords:** maternity rights, psychological factors, transcultural, woman-centered model, healthcare, perception, birth experiences

## Abstract

Maternity rights are perceived and fulfilled differently according to women’s psychosocial characteristics, leading to varying maternal experiences and outcomes. It is necessary to know the impact of cultural context, emotional well-being, and resource availability on the maternal woman’s clinical care experience. The aim is to identify if these factors contribute to disparities in the perception of maternity rights fulfillment in Spain and Colombia. This retrospective observational study focused on women who received maternity-related healthcare in Spain or Colombia. A total of 185 women were included (Spanish = 53; Colombian = 132). Data collected included social and obstetric history, as well as psychological variables such as resilience, positive and negative affect, derailment, and maternity beliefs. The study also assessed women’s knowledge of healthcare rights (MatCODE), perceptions of resource scarcity (MatER), and the fulfillment of maternity rights (FMR). C-section was more prevalent in Colombia, where women also scored higher on maternity beliefs as a sense of life and as a social duty compared to Spanish women. Conversely, FMR was higher in the Spanish context. Colombian women reported lower levels of social support and less involvement in medical decision-making. The FMR was positively correlated with positive affect, MatCODE, and MatER. Predictive modeling identified negative factors for FMR, including giving birth in Colombia (β = −0.30 [−0.58; −0.03]), previous miscarriage (β = −0.32 [−0.54; −0.09]), C-section in the most recent labor (β = −0.46 [−0.54; −0.0]), and higher MatER scores. Positive predictors included gestational age, maternal age, and previous C-section (β = 0.39 [0.11; 0.66]). The perception of the fulfillment of maternity rights depends on socio-healthcare contexts, women’s age, obstetric history, and resources. It is suggested to apply culturally sensitive strategies focused on women’s needs in terms of information, emotional and social support, privacy, and autonomy to manage a positive experience.

## 1. Introduction

Maternity rights encompass a broad spectrum of legal, social, and healthcare provisions aimed at ensuring the well-being of women and families during pregnancy, childbirth, and the postpartum period. These human rights are perceived for the woman depending on its sociocultural and psychosocial factors, leading to disparities in maternal experiences and outcomes. Inequalities in maternity rights fulfillment lead to significant ethical and public health challenges. Ethically, they disrupt human rights and social inequities, which affects women’s confidence and health ([5]). From a public health perspective, it contributes to adverse maternal and infant health outcomes, including increased mortality rates, and impose economic burdens due to higher healthcare costs ([45]). The feeling of unfulfillment with maternity rights may be at the basis of obstetric violence. Obstetric violence refers to gender-based violence that occurs within the healthcare system during pregnancy, childbirth, and postpartum period. It encompasses mistreatment, negligence, abuse, or disrespect of women by healthcare providers, often during childbirth, that violates their rights, dignity, privacy, and autonomy ([2]). According to the Centers for Disease Control and Prevention (CDC; USA), 20% of women reported mistreatment during maternity care. Hispanic (29%) and multiracial (27%) women experienced higher rates of verbal and physical mistreatment. Additionally, 29% reported discrimination, primarily based on age and income ([10]). In the Spanish context, 38.3% of women have experienced obstetrical mistreatment during maternity care ([49]). In Colombia, the data rise close to 70% ([39]).

There is evidence supporting the need for a shift in the model of maternity care, prioritizing nursing and woman-centered care ([47]; [69]; [78]). The new model should empower women, providing information to make decisions (if the life of the woman or fetus is not in danger) and focusing on social support. This model proposes multidimensional care, strengthening the relationship between the professional and the woman ([23]). A theoretical model has been proposed to determine the relationship between women and health institutions ([43]). Later, the model was completed with maternal healthcare obstacles and facilitators in social contexts ([30]), considering itself a more ecological model. However, this approach should extend beyond the birth process, adhering to a biopsychosocial model ([4]), analyzing the social determinants ([83]) and mental health of women ([15]; [94]). A mixed-method study proposes that women’s perception related to childbirth experience influenced mistreatment during maternity care ([50]). Additionally, emotional well-being and social support during pregnancy and postpartum play a crucial role in health and the fulfillment of maternity rights ([74]). Research has shown that resilience, identity, and affect can influence on how women perceive the adequacy of the maternity care ([58]; [73]). A scoping review revealed differences in how women from different cultural backgrounds perceive and experience maternity rights ([71]). In cross-cultural analyses, it has been observed that women from different cultural backgrounds often report varying levels of satisfaction with maternity care, which can be attributed to differences in healthcare systems, social support structures, and cultural expectations. Women in low-resource healthcare settings are more likely to report non-fulfillment of maternity rights, linked to a lack of adequate information, support, and respect during childbirth ([70]). Therefore, the fulfillment of maternity rights is closely associated with the level of resources available, the quality of care provided, and the cultural competence of healthcare providers ([40]).

The current study builds upon this background of research by examining the social, obstetrical, and emotional factors that influence the perception of the fulfillment of maternity rights among women who received healthcare during their maternity in two different socio-health contexts, Spain and Colombia. Both social contexts have differences in the regulatory environments. Spain and Colombia offer access to maternal healthcare under public systems ([16]; [55]). In addition, abortion in Spain is permitted up to 14 weeks of pregnancy and up to 22 weeks if the pregnancy threatens the mother’s health or if there are fetal abnormalities ([59]). Colombia’s laws were expanded to allow abortion up to 24 weeks ([72]). While Spain’s maternal mortality rate is relatively low, Colombia continues to face higher rates, largely due to issues with access and disparities between urban and rural healthcare ([25]; [67]). This is important, considering that both social contexts have humanizing guidelines for motherhood care ([38]; [14]).

Understanding the factors that influence the perception of maternity rights can help inform policies and practices that better support women across diverse cultural settings, ultimately leading to improved maternal health outcomes globally. Furthermore, laws to protect maternity rights must be complemented by health professionals and researchers ([92]). Thus, the hypothesis established was those psychological and social variables in women—such as emotional coping, poor resilience, beliefs regarding motherhood, lack of awareness of maternity rights, and insufficient social support resources—are key determinants of maternal experiences. The present exploratory observational study aims to identify key determinants that contribute to the perception of maternity rights fulfillment, with a focus on the role of obstetrical history, emotional well-being, and resource availability.

## 2. Materials and Methods

### 2.1. Ethical Statement

This study was approved by the Research Ethics Committees (CEI-112-2199, 22 January 2021) of FOSCAL Hospital from Colombia (Santander, Colombia; FOSCAL-06939/2022, 23 September 2022). All women willing to participate were given an online information sheet, describing the aim of the study, and an informed consent form was signed in each case. Data collection was anonymous, and databases were blinded. In addition, this study follows the guidelines for strengthening the reporting of observational studies in epidemiology (STROBE) ([88]) for cross-sectional cohort studies.

### 2.2. Setting Socio-Health Context

This study was carried out in two settings with different socio-health contexts, Spain and Colombia. These settings were identified as being between the distribution of respectful maternity care research, organized by country income category as determined by the World Bank ([82]). According to the review related to the geographical locations of respectful maternity care research ([71]), Spain is a high-income country and Colombia is an upper–middle-income country. The main differences between settings related to care would be that in Spain, the obstetric healthcare system is primarily public, funded by the Spanish government. Spain offers universal health coverage, meaning that women have access to maternal health services regardless of their socioeconomic status. However, they can also choose to have private health insurance. In contrast, Colombia has a mixed healthcare system, composed of both public and private sectors. Access to obstetric care can vary depending on the region and the woman’s socioeconomic level ([31]). Economic and regional disparities can influence the experience and perception of maternity rights. In Spain, midwives play a significant role in the care of pregnancies, being specialized nurses with a wide array of responsibilities designed to support the health and well-being of women throughout pregnancy, childbirth, and postpartum recovery. In Colombia, this role is not institutionalized. Spain has health policies related to maternity rights (i.e., the Patient Autonomy Law) ([38]) and a strategy for assistance in normal childbirth in the Spanish health system ([52]). Although Colombia is progressing with regard to these rights ([14]), challenges remain in the implementation of these policies, especially in less developed areas. Women in low-resource settings may face barriers to accessing adequate information and respectful care during childbirth.

### 2.3. Participants of the Study

Women were selected by non-probabilistic convenience sampling at the discretion of the research team. To reduce potential sampling biases, the representativeness of the sample was implemented by the eligibility criteria. The women were contacted by social media, an adequate technique for recruitment ([27]), using Facebook and Instagram social groups for women in maternity, pregnant, or during the postpartum period, focusing on interests related to health and wellness. Informative videos were used with a direct call-to-action linking to the consent form and survey.

The inclusion criteria of the cohort were women ≥18 years; who had been pregnant in the last 3 years; who had received healthcare for their most recent pregnancy, labor, or postpartum in a tertiary healthcare center in Spain or Colombia; and who had a good understanding of the Spanish language. The exclusion criteria were an inability to read/write in Spanish, home birth, no internet access, and pregnancy at the moment of the study.

During recruitment, 405 women were contacted, but 278 were finally eligible. Then, the inclusion and exclusion criteria were applied. Finally, 185 women met these criteria (Figure 1), being 29.3 years of age during pregnancy. Data were collected from September 2021 to November 2023. In total, 70% of the recruited Spanish women had public healthcare for the entire pregnancy, childbirth, and postpartum, with the 30% receiving private assistance via co-payments. Similar proportions were detected in Colombian women; 70% received health coverage from the Health Insurance Provider (Spanish acronym: EPS), while 30% self-paid for their maternal assistance.

### 2.4. Study Design and Procedure

This study represents a retrospective exploratory observational non-interventionist design with a cross-cultural and woman-centered analysis strategy. A self-administered online tool was prepared using Qualtrics (https://www.qualtrics.com/es/ accessed on 15 July 2021). Firstly, it obtained social and obstetrical variables. Secondly, it collected psychological and perception scales.

The social and obstetric history variables were age (years), education level, working situation, civil status, type of family (mono- vs biparental), number of pregnancies (gravida), number of labors (parity), number of previous history of miscarriages, and number of previous C-sections. Related to the most recent pregnancy were the following: use of assisted reproduction techniques (ART; yes/no), presence of multiple pregnancies (yes/no), type of labor (vaginal/C-section), gestational age (completed weeks), preterm birth (labor <37 weeks; yes/no), and adverse outcomes (yes/no) during pregnancy (e.g., preeclampsia or gestational diabetes), labor (e.g., premature rupture of the membrane or intrapartum hemorrhage), early postpartum (e.g., mastitis or sepsis), fetal (e.g., intrauterine growth restriction) or neonatal (e.g., ventricular hemorrhage or chronic lung disease). The selected variables were chosen based on their direct relevance to the research question and previously explored ([74]).

#### 2.4.1. Psychological Instruments to Explore Emotional Variables

Women responded to the four self-report Spanish-validated psychosocial tools, including:

Resilience Scale. This scale measures the ability of the women to recover from stressful circumstances, considered as a positive personality characteristic that allows women to adapt to adverse situations ([68]). The resilience scale was based on the original scale proposed by Wagnild and Young ([89]), but was a short 14-item version with Likert responses from 1 = “Strongly disagree” to 7 = “Strongly agree”. The higher the score, the greater the woman’s ability to cope with the problems of life. Other studies reported a reliability between 0.79 and 0.91 ([33]; [80]). In the present study, a reliability of 0.91 was reported.

The Positive and Negative Affect Schedule (PANAS). The PANAS is a measure that is made up of two mood scales, one measuring positive affect and the other measuring negative affect ([91]). This scale has 20 items (10 items for positive affect and 10 items for negative affect), which are scored based on 5-point Likert scale ranging from 1 = “Very slightly or not at all” to 5 = “Extremely”. For the positive score, a higher score indicates more positive emotions. For the negative score, a lower score indicates fewer negative emotions ([46]). Previous application of the PANAS obtained a reliability between 0.87 to 0.91 ([20]). In the present study, a 0.81 of reliability was reported.

Derailment Scale. The degree to which women perceive change over time in self and direction constitutes an important individual difference ([63]). Therefore, this instrument assesses the women’s feelings of being temporally discrepant and off-sense, called derailment ([8]; [12]). Derailment was indexed via 10 items with 5-point Likert scale responses from 1 = “Strongly disagree” to 5 = “Strongly agree”. The higher the score on the scale, the greater the woman’s feeling of being derailed. The previous reliability of the derailment scale was between 0.75 and 0.90 ([63], [64]). In the present study, a reliability of 0.77 was reported.

The Maternity Beliefs Scale (MBS). Beliefs about motherhood could determine women’s perceptions of childbirth and the process of adaptation to maternity ([62]). A criticism of the scale, reported by the authors, is that it does not focus solely on birth beliefs. However, this scale may be useful in assessing the generality of attitudes towards childbirth. The MSB identifies beliefs that women have about motherhood, clustered into maternity as a sense of life (MBS—life, 8 items) and maternity as a social duty (MBS—social, 5 items). The higher the score, the higher the woman’s belief in the domain. The previous reliability of the MBS was between 0.83 and 0.93 ([29]). In the present study a reliability of 0.91 was reported.

#### 2.4.2. Perception Scales to Explore Maternity Rights and Resources

Subsequently, the women completed three additional questionnaires related to their knowledge and self-perception focus on their latest pregnancy, childbirth, and early postpartum. These tools were validated by the research group, and they focused on:

The knowledge of obstetric healthcare rights (MatCODE). This questionnaire was designed to assess the knowledge that women have of their healthcare rights during pregnancy, labor, or postpartum ([76]). The MatCODE is a 11-item scale scored in a Likert format from 1 = “Strongly disagree” to 5 = “Strongly agree”. Higher scores in MatCODE would indicate a greater awareness of their healthcare rights. The previous reliability of the MatCODE was 0.94 ([76]). In the present study a 0.95 of reliability was reported.

The perception of resource scarcity (MatER). The MatER was designed to assess the woman’s perception of pregnancy, labor, or early postpartum resources ([76]). The MatER is a 9-item scale scored in a Likert format from 0 = “Never” to 4 = “Always”. A higher score in MatER would indicate a lower perception of resources by the woman. The previous reliability of the MatCODE was 0.78 ([76]). In the present study, a 0.79 of reliability was reported.

The fulfillment perception of maternity rights (FMR). Based on the recommendations of the World Health Organization ([93]), the FMR assesses the perception of the fulfillment of women’s rights to adequate healthcare during maternity. In addition, the FMR has 5 dimensions that cover the following: the perception of the women related to receiving adequate healthcare information (Information, 9 items), concerns related to privacy and confidentiality of health information (Privacy, 6 items), concerns related consent to medical procedures (Consent, 4 items), concerns related to social support during maternity (Support, 3 items), and concerns related to participation and active listening in medical treatment (Participation, 7 items). The items cover the most recent pregnancy, childbirth, and postpartum. The scale ranged from 1 to 4 in a Likert response, in which 0 = “Never” and 4 = “Always”, being the higher FMR score, the higher perception of the right’s fulfillment. The previous reliability of the FMR was between 0.91 to 0.94 ([75]). In the present study a 0.92 of reliability was reported.

### 2.5. Statistical Analysis

Following the theory of the central limit (*n* > 100), the data were described by mean and the standard error of the mean (SEM) of the quantitative variables. For the qualitative variables, the data are summarized as the relative frequency (%) and sample size (*n*). To the univariate analysis was applied an unpaired Student’s *t*-test for the quantitative variables, and a chi-squared test was used in the proportion comparison. The correlation to test the different pattern between socio-health context in FMR and psychological variables were tested by Pearson’s coefficient (ρ) at a 95% confidence interval (95% [CI]).

The multivariate analysis was tested by linear regression models to explain the association between the perception of FMR and the social characteristics, obstetrical health history, and emotional and perceptional psychological variables of the women. The adjusted variables were introduced if they were associated with an error probability (*p*) < 0.1 in the univariate analysis. From the models were extracted the standardized coefficient (β) with 95% CI. In this analysis, no imputation techniques were used for missing values. In all the analyses, a *p*-value (*p*) < 0.05 was considered statistically significant.

The descriptive and inferential analyses were performed using R software within the RStudio interface (version 2022.07.1+554, 2022, R Core Team, Vienna, Austria) with the *rio, dplyr, compareGroups, devtools, psych,* and *lavaan* packages.

## 3. Results

### 3.1. Social and Obstetrical Characteristics

The women’s age showed a trend of being higher in the Spanish cohort than the Colombian one, without differences in other social variables (Table 1). Overall, 1.17 ± 1.2 years had passed since the women’s most recent pregnancy, with more postpartum time elapsing until entering the study in Spanish cohort than the Colombian.

Related to the obstetrical history, the miscarriage rate was significantly higher in the Spanish than in the Colombian context, the inverse of the previous criterion, C-section rate, which was more prevalent in Colombia than in Spain. Additionally, it was more prevalent in Colombia than in Spain that the most recent labor was by C-section. A total of 87.3% of the women stated that their most recent pregnancy was intended, with 4.9% being by ART, 1.6% being multiple pregnancies, and 7.6% being preterm birth. Although prematurity was similar between both contexts, the Colombian cohort had a significantly lower gestational age than the Spanish context. No differences were detected between cohorts in the adverse obstetrical, fetal, or neonatal outcomes (Table 2).

### 3.2. Emotional Variables During the Most Recent Pregnancy and Postpartum

During the most recent pregnancy and postpartum, resilience and affect scales were similar between cohorts. However, women in the Colombian context tended to have higher derailment, and they scored significantly higher in maternity as a sense of life and as a social duty compared to the Spanish context (Table 3).

### 3.3. Perception of Maternity Rights and Resources During the Most Recent Pregnancy, Childbirth and Postpartum

The women’s knowledge of healthcare rights and their perception of resources were similar between cohorts (Figure 2A,B). However, the self-perception of the fulfillment of maternity rights was significantly lower in women who gave birth in Colombia than in women who gave birth in Spain (Figure 2C). In addition, although the perception of fulfillment of rights related to receiving adequate healthcare information (Figure 2D) and to consenting to medical procedures (Figure 2F) were similar, the rights of social support during maternity (Figure 2G) and to participation in medical treatment (Figure 2H) were significantly lower in the Colombian than the Spanish context. Also, the rights related to privacy and confidentiality of health information were somewhat lower in the Colombian cohort (Figure 2E).

### 3.4. Correlations Between Perception of Maternity Rights and Emotional Variables

Resilience did not correlate with the perception of the fulfillment of maternity rights in any case. In the global population, positive emotions correlated significantly and positively with the perception of fulfillment of these rights. Furthermore, women who gave birth in Spain had a significant and negative correlation of negative emotions with the perception of fulfillment of maternity rights. In addition, women in the Colombian context had a significant and positive correlation with positive emotions and the perception of fulfillment of maternity rights. Furthermore, those in the Spanish context had a negative and significant correlation between the fulfillment of maternity rights and derailment and the most significant correlation between the fulfillment of maternity rights and maternity as a social duty (Table 4).

Generally, the knowledge of rights was significantly and positively correlated with the fulfillment of maternity rights. However, this perception seems to be associated with women who gave birth in Spain. Furthermore, perceiving scarcity of resources was significantly and negatively correlated with the perception of these rights. This pattern was observed in both socio-health contexts (Table 4).

Due to showing correlation in the global cohort, positive emotions, the knowledge of maternity rights, and scarcity of resources were introduced into the associative models.

### 3.5. Woman-Centered Model to Explain Their Perception of Fulfillment of Maternity Rights

Overall, Colombian socio-health context (β = −0.30 [–0.58; −0.03]), previous miscarriage (β = −0.32 [−0.54; −0.09]), exposure to C-section in the most recent labor (β = −0.46 [−0.92; −0.0]), and perceiving scarcity of resources (β = −0.03 [−0.05; −0.01]) were negative factors associated with fulfillment of maternity rights. Conversely, positive factors were women’s age (β = 0.02 [0.0; 0.04]), previously exposition to C-section (β = 0.39 [0.11; 0.66]) and increased of gestational age (β = 0.07 [0.0; 0.14]; Figure 3A).

Specifically, negative factors to perceive inadequate healthcare information included previous miscarriages, having a C-section in the most recent labor, beliefs in maternity as a social duty, and a high perception of resource scarcity. On the other hand, factors that increased the perception of fulfillment of this right were older maternal age, increased gestational age, and previous C-section (Figure 3B). Regarding the fulfillment of rights related to privacy and confidentiality, desired for pregnancy was a protective factor (Figure 3C). The perception of fulfillment of rights related to social support decreased when in Colombia context and the women perceived a scarcity of resources (Figure 3D). Most recently, the fulfillment of rights related to participation and active listening in medical treatment decreased in Colombian context, most recent labor by C-section, and the perception of resource scarcity, but increased with previous C-sections and the woman’s knowledge of healthcare rights (Figure 3E).

## 4. Discussion

The main contribution of this work indicates that the perception of maternity rights is significantly influenced by women’s previous experiences and biopsychosocial factors, which strengthen our hypothesis. Globally, the socio-health context of childbirth impacts this perception, particularly in women who give birth in Colombian context and who perceive low respect for their rights, emotional support, and participation during motherhood compared to the Spanish context. In addition, obstetrical history and resource availability would be key determinants. Previous experiences related to motherhood (miscarriage and C-section) can cause an ambivalent condition in women’s perception of fulfillment of rights. Age, gestational age, and knowledge of rights can be protective factors. However, scarcity of resources (personal and practical) was a risk determinant for the perception of vulnerability of rights (Table 5).

The difference in perception and women’s experience related to socio-health contexts could be explained by care-technology accessibility and funding of hospitals ([85]). Health policies may limit the fulfillment of rights due to a lack of investment in resources and non-renovation of humanized care protocols ([37]; [39]). Rural areas show lower odds of timely maternity care than urban locations; thus, increasing the healthcare providers can improve adequate maternity care for Hispanics ([1]). Additionally, in low-income populations, access to quality services decreases, increasing the perception of obstetric violence ([87]). This must be considered, as the woman’s economic situation and co-payments, typically associated with urban areas and greater economic resources, can influence her experience of rights. In both contexts, the women had maternal care in urban hospitals, more than 41% were unemployed, and 30% of the women were attending under self-payment conditions. Therefore, key strategies to enhance maternity care include culturally appropriate interventions, strategic of resources to underserved populations, and women empowerment to control their health and well-being.

Adaptation to motherhood involves changes that require internal and external resources ([3]). Resources can act as a barrier or facilitator in the fulfillment of motherhood rights, not only regarding material/economic resources, but also emotional/affective resources (family, friends, partners, or even work–life balance policies). Our results show that the perception of lack of resources is a barrier in motherhood adaptation. Similarly, Mexican women living in the USA who received less emotional support from families were less likely to seek prenatal care, adopt healthy behaviors (such as avoiding smoking), or feel enthusiastic about their newborns ([22]). Women’s emotions are regulated through social support, which may reduce the fear of childbirth or having a child born with illness ([61]), postpartum depression, as well as protect breastfeeding, and increase self-efficacy during motherhood ([6]; [24]). Furthermore, material resources are important to be a comfortable and resourceful birth environment associated with positive experiences and improve birth outcomes ([11]).

A valuable resource is information, which involves gathering holistic information during medical care, understanding women’s experiences, and providing advice, positive feedback, or information about the pregnancy process and health conditions. Among pregnant women, informational support can lead stressful situations ([28]) or decrease stress and anxiety. Women with satisfactory information during maternity have effectively coped with motherhood changes and have adhered to breastfeeding ([53]; [66]). Other research highlights cultural barriers to accessing resources, such as difficulties understanding, and misalignment between cultural customs and biomedical care ([7]; [34]). Thus, our data reveal that empowering women during motherhood with resources emphasizes their perceived fulfillment of rights to information, and participation.

Aged women tend to have more experience with healthcare systems. With age, many women develop greater security and self-esteem ([57]) and are better informed, empowered in decisions, and assertive demanding the fulfillment of rights ([87]). Similarly, gestational age facilitated the feeling of fulfillment of rights. As the pregnancy progresses, women could receive more medical visits and follow-up and information ([84]). Both women’s age and gestational age are factors that may be associated with greater awareness of the experience of motherhood to demand adequate information.

Women think that their privacy and expectations are harmed by being subjected to more technical healthcare ([36]). According to our models, undergoing a C-section in the most recent birth results in less fulfillment’s rights, but, optimistically, the C-section experience can change the women’s mental schemas and prepare her for future pregnancies, since previous C-sections can positively influence this perception. It is necessary to consider that the C-section is an invasive technique, with long recovery periods that could be complicated and in which greater socio-health care would be necessary. In Canadian women, the main predictor of a negative birth experience was the C-section ([77]). In Swedish women, emergency, but not elective, C-section was associated with a negative experience of labor ([90]). In Peruvian ([86]), Chilean ([9]), Italian ([51]) and Portuguese ([81]) women, the following were identified as predictors of negative birth experience: lack of respectful maternity care and privacy, disrespect during childbirth, lack of continuity of care, and poor communication with healthcare providers. However, the lack of control during childbirth and C-section continues to be indicators of a bad experience during maternity. Previous experiences can influence how to process and remember information. According to social cognition theories, experiences contribute to the development of mental schemas that organize information about the world ([26]). If a woman has experienced a previous C-section in a health context where maternity rights were respected, she would be more likely to develop a positive schema regarding these rights. Women with previous C-sections may adjust expectations, giving control over the consequences of subsequent labor. A woman who experiences cognitive dissonance when experiences and beliefs about rights are in conflict may also adjust her perception of the rights to align closely with expectations and provide adaptation. According to our data, previous C-section experience and knowledge of maternity rights were positive predictors for perceived maternity rights fulfillment, particularly important for the adequate information and participation in healthcare decisions. Moreover, the perception of rights is not affected by the feeling of derailment, confirming the cognitive state of adjustment.

According to the ecological model, previous affective events for woman (such as miscarriage) can influence behavior and development ([54]). Miscarriages are associated with mental health consequences depending on the cultural and individual characteristics ([18]), and they are frequently associated with high levels of distress, anxiety, and depression ([19]). A prospective study showed the need for psychiatric treatment within 6 months of the labor of their first live birth in women with a history of miscarriage ([65]). According to our data, previous experience of miscarriage was a negative factor in reducing the women’s perception of rights, especially information. However, findings from the USA indicated no association between experiences of miscarriage and maternity feelings ([41]), other data with Europe women reported that abortion was a risk of negative feeling related to the maternity process ([90]), post-traumatic stress, anxiety and depression ([35]). Therefore, identifying women at risk, and designing psychosocial interventions can reduce adverse effects not only in the months after the miscarriage but also during future pregnancies. Globally, these data show that previous experiences in the woman’s obstetric history are key to addressing patient satisfaction and confidence during motherhood.

Finally, it is necessary to highlight the influence of sociocultural beliefs and practices on maternal healthcare outcomes. There are described barriers related to beliefs, particularly by male partners, and the society within which women live ([79]). In many Western societies, motherhood is a central goal of women’s lives, linked to being responsible for procreation and caring for children, putting the needs of the children before their own ([44]; [56]) and straining their desire for professional development ([13]). It has shown that societies can cause greater indecision in women’s feelings of maternity ([21]). Other authors have shown that women’s desires to be a “perfect” mother are related to increased maternal guilt, lower self-efficacy, and higher stress levels ([48]). The present data added that when maternity is perceived as a duty to the society into which women are integrated, this can be harmful. The results suggest that seeking appropriate maternity care with respect for women’s autonomy from their social roles, promoting conscious choices, and the full support of partners and resources are rights-based issues.

### Strengths and Limitations

Previous research has focused on analyzing variables in health professionals and institutions that influence the vulnerating of women’s rights during maternity care. In turn, this study reports factors that should be integrated into the comprehensive assessment and intervention processes during pregnancy, childbirth, and the postpartum period. Women with limited resources, previous miscarriage, and C-section may require greater support to have a positive perception of healthcare. Thus, complications that arise from negative experiences could be prevented ([74]).

Although we did not find major differences in the sociodemographic variables between both socio-health contexts, these data do not accurately determine the income level of the family unit. Knowing the discrepancies between health policies in both contexts, it would be interesting to be able to explore these differences by salary level. On the other hand, since health strategies can be very discrepant and following the country income category by the World Bank ([71]), it would be interesting to explore the fulfillment of rights in contexts classified as low-income countries and its relationship with other measures such as autonomy, childbirth experience questionnaire ([60]), and the respectful maternity care ([32]). Another research gap would be focused on populations with vulnerable variables. Previous data showed health disparities in maternity care in women who are at risk for pregnancy-related mortality ([17]). Specifically, women who are racialized, younger, not married, with low educational level, and who receiving governmental benefit plans ([42]) are candidates for delaying healthcare.

In addition, considering the methodology, the sample size distribution and asymmetry in the subletting can be a limitation. However, although age among women tended to be significant, the rest of the social variables were homogeneous. The significant obstetric characteristics were used to adjust the models. Therefore, robust statistical methods were employed to mitigate its impact and ensure the validity of comparative analyses.

## 5. Conclusions

Socio-healthcare contexts play a crucial role in shaping women’s perceptions of maternity rights. In this study, Colombian women reported feeling less respected and supported in maternity care compared to their Spanish counterparts. Factors such as age, gestational stage, and previous obstetrical experiences influence these perceptions. Empowering women through improved access to information, emotional support, and resources is essential to enhance their sense of rights fulfillment during maternity. Together with social context, meeting women’s expectations is necessary to prioritize psychological support during pregnancy and postpartum to reduce anxiety, depression, and frustration, and support better adaptation to motherhood. Policymakers should prioritize culturally sensitive healthcare practices, allocate resources effectively, and create maternity care environments that uphold women’s rights and encourage active participation, especially for vulnerable populations.

## Figures and Tables

**Figure 1 ejihpe-15-00010-f001:**
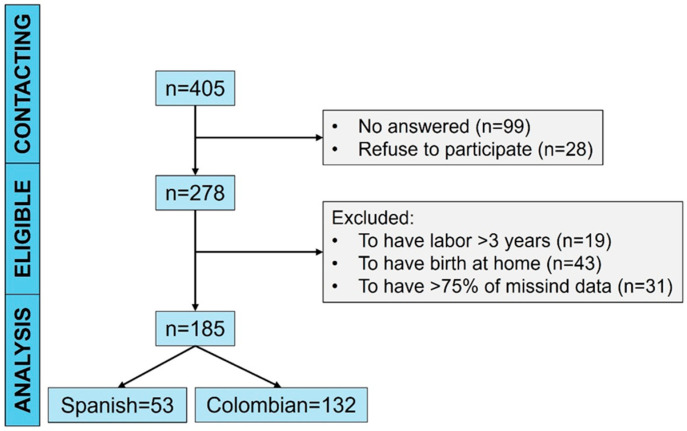
Flow diagram of women participating in the study. The analysis stage was women who gave birth in a Spanish or Colombian socio-health context. Sample size (n). Adapted from STROBE guidelines ([88]).

**Figure 2 ejihpe-15-00010-f002:**
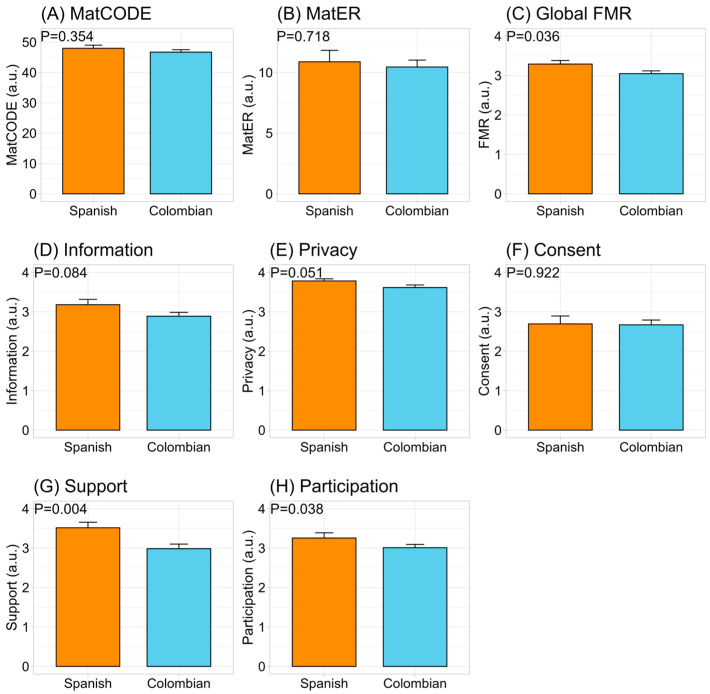
Perception of maternity rights and resources during the most recent pregnancy, childbirth, and postpartum between countries of labor. Data show mean ± standard error of mean (SEM). The *p*-value (*p*) was extracted from an unpaired Student’s *t* test. The women’s knowledge of healthcare rights (MatCODE); the perception of resource scarcity (MatER); the fulfillment of maternity rights (FMR), and its dimensions (information, privacy, consent, support and participation).

**Figure 3 ejihpe-15-00010-f003:**
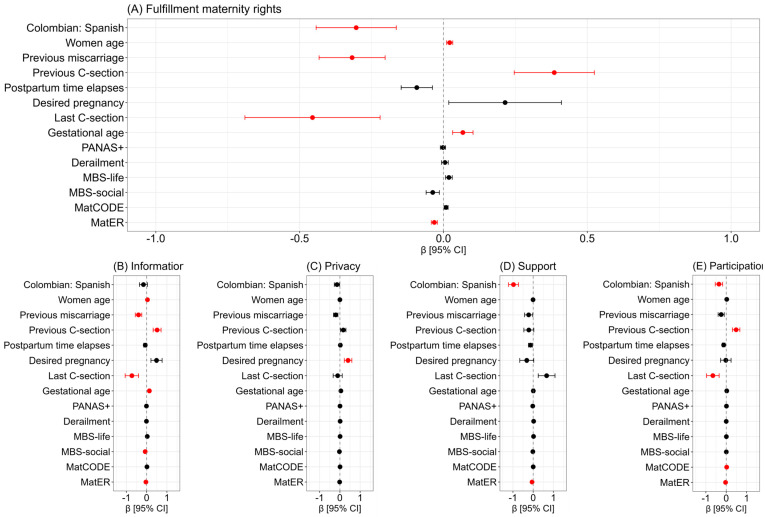
The models of perception of fulfillment of maternity rights. Data show the adjusted and standardized coefficients (β) with 95% confidence intervals [CI] obtained from a linear regression model. The red dot means significant association (*p* < 0.05) and the black dots indicate non-significant association. The global fulfillment of maternity rights (**A**), and its significant dimensions related to rights (Colombian vs Spanish; (**B**) information, (**C**) privacy, (**D**) support and (**E**) participation). The Positive Affect Schedule (PANAS+); the Maternity Beliefs Scale, clustered in maternity as a sense of life (MBS—life) and maternity as a social duty (MBS—social); the women’s knowledge of healthcare rights (MatCODE); the perception of resource scarcity (MatER).

**Table 1 ejihpe-15-00010-t001:** Social variables compared to the country of labor.

	Total (*n* = 185)	Spanish (*n* = 53)	Colombian (*n* = 132)	*p*
Women’s age (years)	29.3 ± 6.3	30.6 ± 5.8	28.8 ± 6.5	0.054
Educational level				
Primary school	4.9% (9)	1.9% (1)	6.1% (8)	0.625
Secondary school	50.8% (94)	52.8% (28)	50.0% (66)	
University	44.3% (82)	45.3% (24)	43.9% (58)	
Working situation				
Employed	58.9% (109)	52.8% (28)	61.4% (81)	0.367
Unemployed	41.1% (76)	47.2% (25)	38.6% (51)	
Civil status				
Single	18.9% (35)	18.9% (10)	18.9% (25)	>0.999
Married	81.1% (150)	81.1% (43)	81.1% (107)	
Biparental family	84.3% (156)	90.6% (48)	81.8% (108)	0.209

Data show mean ± standard error of mean (SEM) in quantitative variables and relative frequency (%) and sample size (*n*) in qualitative variables. The *p*-value (*p*) was extracted via unpaired Student’s *t* test or chi-squared test according to type of variable.

**Table 2 ejihpe-15-00010-t002:** Obstetrical characteristics during the most recent pregnancy and postpartum compared to the country of labor.

	Total (*n* = 185)	Spanish (*n* = 53)	Colombia (*n* = 132)	*p*
Gravida	1.7 ± 1.0	1.7 ± 0.8	1.8 ± 1.1	0.441
Parity	1.4 ± 0.9	1.4 ± 0.6	1.5 ± 1.0	0.312
Previous miscarriage	0.2 ± 0.5	0.4 ± 0.7	0.1 ± 0.4	0.020
Previous labor by C-section	0.7 ± 0.8	0.3 ± 0.6	0.8 ± 0.9	<0.001
Postpartum time elapsed (years)	1.17 ± 1.20	1.57 ± 1.25	1.01 ± 1.14	0.006
Assisted reproduction techniques	4.9% (9)	9.4% (5)	3.0% (4)	0.122
Multiple pregnancy in the recent gestation	1.6% (3)	3.8% (2)	0.8% (1)	0.198
Desired most recent pregnancy	87.3% (144)	100% (53)	81.2% (91)	0.002
Most recent labor by C-section	46.5% (86)	22.6% (12)	56.1% (74)	<0.001
Gestational age (completed weeks)	38.7 ± 1.7	39.1 ± 2.0	38.5 ± 1.6	0.034
Preterm birth	7.6% (14)	11.3% (6)	6.1% (8)	0.230
Obstetrical complications	31.4% (58)	88.7% (47)	93.9% (124)	0.968
Fetal complications	17.8% (33)	20.8% (11)	16.7% (22)	0.657
Labor complications	20.5% (38)	28.3% (15)	17.4% (23)	0.146
Postpartum complications	17.8% (33)	17.0% (9)	18.2% (24)	>0.999
Neonatal complications during labor	13.5% (25)	17.0% (9)	12.1% (16)	0.525
Neonatal complications during postpartum	10.3% (19)	9.4% (5)	10.6% (14)	>0.999

Data show mean ± standard error of mean (SEM) in quantitative variables and relative frequency (%) and sample size (*n*) in qualitative variables. The *p*-value (*p*) was extracted from an unpaired Student’s *t* test or chi-squared test according to type of variable.

**Table 3 ejihpe-15-00010-t003:** Emotional variables during the most recent pregnancy and postpartum compared to the country of labor.

	Total (*n* = 185)	Spanish (*n* = 53)	Colombia (*n* = 132)	*p*
Resilience	82.1 ± 13.2	83.0 ± 9.4	81.7 ± 14.5	0.469
PANAS—positive	36.7 ± 7.6	37.7 ± 7.1	36.2 ± 7.8	0.252
PANAS—negative	23.8 ± 8.7	23.3 ± 9.2	24.0 ± 8.5	0.644
Derailment	20.4 ± 5.4	19.4 ± 5.2	20.9 ± 5.5	0.055
MBS—life	17.3 ± 7.4	14.8 ± 6.4	18.4 ± 7.6	0.003
MBS—social	8.0 ± 3.7	6.7 ± 2.6	8.5 ± 3.9	0.001

Data show mean ± standard error of mean (SEM). The *p*-value (*p*) was extracted from an unpaired Student’s *t* test. The Positive and Negative Affect Schedule (PANAS); the Maternity Beliefs Scale, clustered in maternity as a sense of life (MBS—life) and maternity as a social duty (MBS—social).

**Table 4 ejihpe-15-00010-t004:** Correlations between the perception of women related to the fulfillment of maternity rights and significant emotional variables between country of labor.

	Total	Spanish	Colombian
Resilience	0.11 [−0.06; 0.27] *p* = 0.220	0.17 [−0.11; 0.43] *p* = 0.235	0.09 [−0.13; 0.30] *p* = 0.433
PANAS—positive	0.17 [0.00; 0.33] *p* = 0.047	0.17 [−0.11; 0.43] *p* = 0.231	0.23 [0.04; 0.40] *p* = 0.016
PANAS—negative	−0.10 [−0.26; 0.07] *p* = 0.260	−0.40 [−0.61; −0.14] *p* = 0.004	0.04 [−0.17; 0.25] *p* = 0.699
Derailment	−0.09 [−0.25; 0.08] *p* = 0.304	−0.32 [−0.55; −0.04] *p* = 0.023	0.02 [−0.20; 0.23] *p* = 0.883
MBS—life	0.02 [−0.15; 0.19] *p* = 0.808	−0.10 [−0.37; 0.18] *p* = 0.493	0.12 [−0.10; 0.32] *p* = 0.289
MBS—social	−0.08 [−0.24; 0.09] *p* = 0.372	−0.24 [−0.49; 0.04] *p* = 0.089	0.01 [−0.21; 0.22] *p* = 0.937
MatCODE	0.19 [0.03; 0.35] *p* = 0.027	0.24 [−0.04; 0.48] *p* = 0.095	0.16 [−0.06; 0.36] *p* = 0.146
MatER	−0.31 [−0.46; −0.16] *p* < 0.001	−0.42 [−0.62; −0.15] *p* = 0.003	−0.29 [−0.48; −0.08] *p* = 0.007

Data shows correlation coefficient and 95% confidence interval [CI]. The *p*-value (*p*) was extracted from Pearson’s correlation. The Positive and Negative Affect Schedule (PANAS); The Maternity Beliefs Scale, clustered in maternity as a sense of life (MBS—life) and maternity as a social duty (MBS—social); The women’s knowledge of healthcare rights (MatCODE); the perception of resource scarcity (MatER).

**Table 5 ejihpe-15-00010-t005:** Main results related to the socio-health context of childbirth impact on fulfillment of maternity rights (FMR).

	Global FMR	FMR Information	FMR Support	FMR Participation
Colombian socio-health context	−			−
Women’s age	+	+		
Previous miscarriage	−	−		
Previous C-section	+	+		+
Desired pregnancy			+	
Most recent C-section	−	−		−
Gestational age	+	+		
Maternity beliefs as a social duty		−		
Knowledge of maternity rights				+
Scarcity of resources	−	−		−

Risk factor due to decreased FMR (−); protective factor due to increased FMR (**+**), being adjusted by the postpartum time elapsed to respond to the questionnaires, positive affect, derailment and maternity beliefs as a sense of life.

## Data Availability

The original contributions presented in the study are included in the article, further inquiries can be directed to the corresponding author.

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
