# Peer review of "Analysis of Maternity Rights Perception: Impact of Maternal Care in Diverse Socio-Health Contexts"

_ejihpe, 2025, doi:10.3390/ejihpe15020010_

Round 1
Reviewer 1 Report
Comments and Suggestions for Authors
The aim of the study is to identify key determinants that contribute to the perception of maternity rights fulfillment, focusing on the role of obstetrical history, emotional well-being, and resource availability in two distinct socio-health contexts—Spain and Colombia.
The paper is interesting and well-written, however, there are some suggestions to improve it:
Introduction:
1. Elaborate on why disparities in maternity rights fulfillment are problematic, both from ethical and public health perspectives.
2. Add research hypothesis.
Methods:
1. State sampling method (convenience?)
2. Elaborate on the use of social media for recruitment (e.g., platforms used, targeting strategies).
Discussion:
1. I suggest the authors begin this section with a concise recap of the most important results and not with the importance/innovation.
2. Suggest actionable strategies for improving maternity care, such as developing culturally tailored interventions or improving resource allocation in underserved areas, and emphasize the importance of empowering women.
Conclusion:
Add future research directions.
Author Response
The aim of the study is to identify key determinants that contribute to the perception of maternity rights fulfillment, focusing on the role of obstetrical history, emotional well-being, and resource availability in two distinct socio-health contexts—Spain and Colombia.
The paper is interesting and well-written, however, there are some suggestions to improve it:
Response: Thank you for your time reviewing our manuscript. Your suggestions were considered and responded below.
Introduction:
- Elaborate on why disparities in maternity rights fulfillment are problematic, both from ethical and public health perspectives.
- Add research hypothesis.
Response: The introduction has been expanded according to the reviewer's suggestions (lines 47-51). Additionally, a hypothesis was added in the main text (lines 115-118).
Methods:
- State sampling method (convenience?)
- Elaborate on the use of social media for recruitment (e.g., platforms used, targeting strategies).
Response: Thank you for these recommendations that were implemented. In addition, the use of social media for recruitment has been elaborated (lines 162-167). In addition, although the sampling was non-probabilistic for convenience, the eligibility criteria helped reduce potential sampling biases.
Discussion:
- I suggest the authors begin this section with a concise recap of the most important results and not with the importance/innovation.
- Suggest actionable strategies for improving maternity care, such as developing culturally tailored interventions or improving resource allocation in underserved areas and emphasize the importance of empowering women.
Response: The discussion has been reformulated to adapt to the reviewer´s recommendations, starting with the most notable achievements of the data. New text has been proposed with healthcare strategies in unfavorable contexts and the empowerment of women with resources such as education (lines 538-541).
Conclusion: Add future research directions.
Response: The discussion has been reformulated to implement future directions (lines 436-439).
Reviewer 2 Report
Comments and Suggestions for Authors
This manuscript addresses a significant research domain pertaining to quality and equity in women's healthcare delivery. However, several methodological and contextual modifications would strengthen the paper:
The manuscript would benefit from a comprehensive overview of maternal rights frameworks in both Spain and Colombia, particularly given its international readership. This contextual foundation is essential for understanding the regulatory and policy environments that influence healthcare delivery in these jurisdictions.
A methodological limitation lies in the sample size and distribution. The current sample demonstrates considerable asymmetry, with Colombian participants (n=132) more than doubling their Spanish counterparts (n=53), potentially compromising the robustness of comparative analyses.
The discussion section requires restructuring. It should begin by clearly stating what the study found, followed by an explicit articulation of how these results extend or challenge existing knowledge in the field. The current comparative analysis relies heavily on benchmarking against highly developed nations (USA, Canada, Sweden), which may not offer the most relevant points of comparison. Including studies from demographically and economically comparable contexts would provide more meaningful insights.
The manuscript's contribution to the existing body of knowledge requires clarification. While the research topic is undoubtedly valuable, the specific knowledge gap being addressed remains undefined. The study's innovative aspects and unique contributions to the field need to be more explicitly delineated.
Author Response
This manuscript addresses a significant research domain pertaining to quality and equity in women's healthcare delivery. However, several methodological and contextual modifications would strengthen the paper:
Response: Thank you for your time reviewing our manuscript. Your suggestions were considered and responded below.
The manuscript would benefit from a comprehensive overview of maternal rights frameworks in both Spain and Colombia, particularly given its international readership. This contextual foundation is essential for understanding the regulatory and policy environments that influence healthcare delivery in these jurisdictions.
Response: The introduction has been expanded with the frameworks of maternal rights to healthcare in Spain and Colombia (lines 96-105).
A methodological limitation lies in the sample size and distribution. The current sample demonstrates considerable asymmetry, with Colombian participants (n=132) more than doubling their Spanish counterparts (n=53), potentially compromising the robustness of comparative analyses.
Response: This is a great consideration and for this reason, robust statistical analysis methodologies were used. However, this consideration was raised in the limitations of the study (lines 556-560).
The discussion section requires restructuring. It should begin by clearly stating what the study found, followed by an explicit articulation of how these results extend or challenge existing knowledge in the field. The current comparative analysis relies heavily on benchmarking against highly developed nations (USA, Canada, Sweden), which may not offer the most relevant points of comparison. Including studies from demographically and economically comparable contexts would provide more meaningful insights.
Response: The discussion has been reformulated to adapt to the reviewer´s recommendations, starting with the most notable achievements of the data. In addition, other comparable contextual areas to the Spanish and Colombian have been exemplified (lines 483-489).
The manuscript's contribution to the existing body of knowledge requires clarification. While the research topic is undoubtedly valuable, the specific knowledge gap being addressed remains undefined. The study's innovative aspects and unique contributions to the field need to be more explicitly delineated.
Response: The contribution to the field of knowledge has been intensified and clarified throughout the main text.
Reviewer 3 Report
Comments and Suggestions for Authors
This paper presents a very relevant topic for different contexts, for various reasons: the medicalisation of society, structural gender inequalities and the institutionalisation of birth. It presents innovative and wide-ranging objectives and methodology and fulfils its aims, presenting good methodological support, an interesting discussion and valid recommendations.
Nevertheless, I think the document could benefit from:
1. Better organisation of the abstract, which presents the methodology and results in an unclear way
2. A more in-depth analysis of the theoretical considerations presented in the framework of the article. The literature review addresses issues that are relevant to the work to be done, but are not always sufficiently developed (e.g. the theoretical model mentioned in line 54, but not developed; concepts addressed in the discussion, but not in the framework, such as ‘obstetric violence’, ‘ecological model’, or ‘intensive mothering roles’).
3. A better explanation of some statements or options, such as:
- the reference to the role of midwives in Spain (line 106),
- the reference to the recruitment of participants for the study (line 115),
- the justification for the choice of variables to characterise the participants (from line 139 onwards)
- some criticism of the scales used, particularly the MBS
- some reflection on the position of the authors, centred mainly on the context of the Global North, without a clear discussion of the bias brought about by the theoretical background adopted and the way in which it was chosen. For example, there is no mention of the probable bias of the sample, due to the way it was selected, which would tend to bring out some homogeneity in the characteristics of the women from the two countries analysed, which in fact contain a lot of comparative and internal heterogeneity and diversity.
4. In the conclusions, some of the recommendations are reductive (lines 468 and 469), as they only consider psychological support for women and not the whole context in which they live.
I believe that these observations do not jeopardise the quality of the article, which deserves to be published, as an important contribution to scientific knowledge.
Author Response
This paper presents a very relevant topic for different contexts, for various reasons: the medicalization of society, structural gender inequalities and the institutionalization of birth. It presents innovative and wide-ranging objectives and methodology and fulfils its aims, presenting good methodological support, an interesting discussion and valid recommendations.
Response: Thank you for your time reviewing our manuscript. Your suggestions were considered and responded below.
Nevertheless, I think the document could benefit from: Better organization of the abstract, which presents the methodology and results in an unclear way.
Response: The abstract was restructured.
- A more in-depth analysis of the theoretical considerations presented in the framework of the article. The literature review addresses issues that are relevant to the work to be done but are not always sufficiently developed (e.g. the theoretical model mentioned in line 54, but not developed; concepts addressed in the discussion, but not in the framework, such as ‘obstetric violence’, ‘ecological model’, or ‘intensive mothering roles’).
Response: Thank you for these considerations, which they were clarified in the background and modified the sentences to be more clear.
- A better explanation of some statements or options, such as:
- the reference to the role of midwives in Spain (line 106),
- the reference to the recruitment of participants for the study (line 115),
- the justification for the choice of variables to characterize the participants (from line 139 onwards)
- some criticism of the scales used, particularly the MBS
Response: The explanations of these recommendations were implemented in the main text.
- some reflection on the position of the authors, centered mainly on the context of the Global North, without a clear discussion of the bias brought about by the theoretical background adopted and the way in which it was chosen. For example, there is no mention of the probable bias of the sample, due to the way it was selected, which would tend to bring out some homogeneity in the characteristics of the women from the two countries analyzed, which in fact contain a lot of comparative and internal heterogeneity and diversity.
Response: This is a great consideration and for this reason, robust statistical analysis methodologies were used. However, this consideration was raised in the limitations of the study (lines 556-560).
- In the conclusions, some of the recommendations are reductive (lines 468 and 469), as they only consider psychological support for women and not the whole context in which they live. I believe that these observations do not jeopardize the quality of the article, which deserves to be published, as an important contribution to scientific knowledge.
Response: Thanks for your recommendation. The conclusions have been reformulated to be adapted to the psychological and social context of maternal care.
Round 2
Reviewer 2 Report
Comments and Suggestions for Authors
The manuscript has been revised to address all previous comments in a satisfactory manner. However, I recommend several minor linguistic revisions to enhance clarity and precision:
1. Page 11, Discussion section opening:
Replace "The main finding of this work" with "The main contribution of this work"
2. Page 12, line 436:
Revise to "Therefore, we recommend key strategies to enhance maternity care that will include the creation of culturally appropriate interventions..."
3. Page 15, line 568:
Modify "would be necessary to assistance together with the social..." to "would be necessary to assist together with the social..."
Following these linguistic refinements, I consider the manuscript suitable for publication.
I wish the authors success with their publication.
Author Response
The manuscript has been revised to address all previous comments in a satisfactory manner. However, I recommend several minor linguistic revisions to enhance clarity and precision:
- Page 11, Discussion section opening: Replace "The main finding of this work" with "The main contribution of this work"
- Page 12, line 436: Revise to "Therefore, we recommend key strategies to enhance maternity care that will include the creation of culturally appropriate interventions..."
- Page 15, line 568: Modify "would be necessary to assistance together with the social..." to "would be necessary to assist together with the social..."
Following these linguistic refinements, I consider the manuscript suitable for publication. I wish the authors success with their publication.
Response: Thank you for reviewing our manuscript and give us these recommendations to improve the readability of the manuscript. All their recommendations were applied in the text.